# Modelling of Carotenoids Content in Red Clover Sprouts Using Light of Different Wavelength and Pulsed Electric Field

**Ilona Gałązka-Czarnecka** [1,*] , **Ewa Korzeniewska** [2,*] , **Andrzej Czarnecki** [1], **Paweł Kiełbasa** [3] **and Tomasz Dróżdż** [3]

1. Faculty of Biotechnology and Food Sciences, Lodz University of Technology, Stefanowskiego 4/10, 90-924 Łódź, Poland; andrzej.czarnecki@p.lodz.pl
2. Institute of Electrical Engineering Systems, Lodz University of Technology, Stefanowskiego 18/22, 90-924 Łódź, Poland
3. Faculty of Production and Power Engineering, University of Agriculture in Krakow, ul. Balicka 116 B, 30-149 Kraków, Poland; pawel.kielbasa@urk.edu.pl (P.K.); tomasz.drozdz@office.urk.edu.pl (T.D.)
* Correspondence: ilona.czarnecka@p.lodz.pl (I.G.-C.); ewa.korzeniewska@p.lodz.pl (E.K.); Tel.: +48-42-631-34-63 (I.G.-C.)

**Abstract:** The paper presents the results of influence the light of different wavelengths and pulsed electric fields on the content of carotenoids. Seeds germination was carried out in a climatic chamber with phytotron system. The experiment was carried out under seven growing conditions differing in light-emitting diode (LED) wavelengths and using pulsed electric fields (PEFs) with different strength applied before sowing. Cultivation of the sprouts was carried out for seven days at relative humidity 80% and 20 ± 1 °C. Different light wavelengths were used during cultivation: white light (380–780 nm), UVA (340 nm), blue (440 nm), and red (630 nm). In addition, the pulsed electric field (PEF) with three values of strength equal to 1, 2.5 and 5 kV/cm, respectively, was applied to three series of sprouts before sowing. Sprouts treated with the PEF were grown under white light (380–780 nm). The light exposure time for all experimental series of sprouts was 12/12 h (12 h light, 12 h dark for seven days). Lutein is the dominant carotenoid in germinating red clover seeds, the content of which varies from 743 mg/kg in sprouts grown in red light, 862 mg/kg in sprouts grown in UVA, to 888 mg/kg in sprouts grown in blue light. Blue light in the cultivation of red clover sprouts had the most beneficial effect on the increase of carotenoids content and amounted to 42% in β-carotene, 19% in lutein, and 14% in zeaxanthin. It confirms that modelling the content of carotenoids is possible using UVA and blue light (440 nm) during seed cultivation. An increase in the content of β-carotene and lutein in red clover sprouts was obtained in comparison to the test with white light without PEF pre-treatment, respectively by 8.5% and 6%. At the same time a 3.3% decrease in the content of zeaxanthin was observed. Therefore, it can be concluded that PEF pre-treatment may increase mainly the content of β-carotene.

**Keywords:** carotenoids; traditional food; light-emitting diodes; pulsed electric field; lutein; zeaxanthin; β-carotene

## 1. Introduction

Carotenoids play a key role in human health and should therefore be a regular part of the daily diet. It is important that they are delivered from various sources. Several carotenoids, including β- carotene, β-cryptoxanthin, and α-carotene are classified as provitamin A. Carotenoids other than provitamin A are also important in the human diet, because their high intake is correlated with a

lower risk of developing chronic degenerative diseases including age-related, with special attention to age-related macular degeneration (AMD), cardiovascular diseases and some types of cancer [1–3].

Carotenoids are pigments soluble in lipids. Carotenoids are isoprenoid metabolites synthesized by all photosynthetic organisms (including algae, plants and even cyanobacteria) and some non-photosynthetic organisms such as archaea, fungi, bacteria or animals. Carotenoids are in over 1100 naturally occurring compounds that give colour to many edible parts of plants and flowers from yellow, through to orange and red. In addition, carotenoids can be cleaved to produce compounds with roles as growth regulators, such as abscisic acid (ABA) and strigolactones, as well as other bioactive molecules.

Carotenoids are found in many plant products. They are components of supplements and are also an additive to feed (e.g., lutein, zeaxanthin, β-cryptoxanthin, α-xanthine) in order to obtain the right colour of, for example, farmed fish eggs [3,4].

The literature on the subject has shown that, except for plants stained orange or red, what is associated with the colour of these compounds (e.g., papaya, carrots, peppers, also green plants such as sprouts) is a very good source of carotenoids [3]. In addition, sprouts belong to low-processed food. Their cultivation is fast (several days), easy and relatively cheap. However, there are no reports on the content of carotenoids in red clover sprouts. Commonly, carotenoids are associated with products that are from yellow to red, but these compounds are also commonly found in products containing chlorophylls. Carotenoids in chloroplasts help to absorb an excess of energy and dissipate it in the form of heat. In photosynthesis, carotenoids help to absorb light, but also play an important role in getting rid of solar energy excess. When a leaf is exposed to the full sun, it receives a large portion of energy. If this energy is not properly managed, it can destroy elements essential for photosynthesis. The seed germs are an interesting product containing carotenoids, obtained during just a few days of their cultivation. The germ is called an embryo that has pierced through the seed coat and developed a system (root and cotyledon) for self-feeding.

The European Union (EU) Commission Regulation 208/2013 from 11 March 2013 defines sprouts as a product obtained as a result of germination of seeds and their development in water or other carriers, which can be collected before the formation of proper leaves and intended for consumption as a whole, including seeds [5]. Sprouted seeds are one of the most nutritious and tastiest types of food in the world. Most sprouts can be eaten fresh. They are an addition or the base of salads, they are also suitable for many snacks and dishes, such as soups, dressings, dips and cocktails. Sprouts can enrich the nutritional value of many dishes, such as pizza, casseroles, croquettes, burgers, meat dishes, etc. Currently, the use of germinated seeds is unlimited and original desserts, smoothies and even sweets are prepared.

Sprouts in some especially Asian cuisines have been consumed for a long time and in Europe and the US they occupy an important place in vegetarian diets. Sprouted seeds, due to their composition, can be called a health bomb, because they contain nutrients and many compounds that have a positive effect on human health; they are primarily antioxidant compounds, including plant dyes such as carotenoids. The advantages of germinated seeds are primarily their nutritional and taste qualities, as well as their growing availability and diversity on the market. Seed cultivation can be carried out throughout the year using various solutions. Modern crops can be grown in phytotron (climatic) chambers with an automatic irrigation and temperature regulating system and additional lighting. In addition, assimilation lighting is one of the most important factors of plants grown in such conditions. Unlike photosynthesis, photoreaction is a more qualitative reaction and it depends on the wavelength. Thanks to this it is possible to stimulate photophysiological processes in the plant, affecting the content of bioactive compounds and obtaining their more often favourable composition.

To produce young edible seedlings (sprouts), it is possible to use the seeds of plants belonging, among others, to the *Fabaceae* family (bean beetles). The red clover (*Trifolium pratense L*) belongs to this family. Sprouted red clover seeds have strong antioxidant properties, mainly due to the high content of bioactive compounds. Studies have shown that they also contain a favourable qualitative

and quantitative composition of phytoestrogenic compounds (isoflavones such as daidzein, genistein, formononetin and biochanin A), which have a beneficial effect on the human body. Sprouted red clover seeds can be included in the daily diet. In addition, light-emitting diodes (LEDs) open the possibility of their use for plant growth in a closed environment [6]. More advantageous and more efficient light sources are based on light-emitting diodes (LEDs) due to their advantages, such as light emission in a narrow spectrum band, high efficiency compared to traditional lamps, low voltage operation, photosynthetic regulation, photosynthetic photon flux density (PPFD) and low heat emission [7].

Plant growth, including germ and the profile of biologically active compounds, depends on the genotype, type of exposure (monochrome, combined or white light), its intensity and time [8]. The content of vitamins and microelements increases, and anti-nutritional components, such as trypsin inhibitors, are removed during germination as a result of intensive metabolism. It makes the germs safe for human consumption. The enzymes are activated in seed during germination, including amylolytic, proteolytic and lipolytic ones. Their activity favourably changes the composition of germinated seeds. Starch, proteins and fats are broken down, becoming a source of energy and substrates for the synthesis of new substances. Literature data [9] indicate the relationship between the content of antioxidant compounds in sprouts and their growth conditions (i.e., seed location, temperature and humidity). Seed cultivation can be carried out using many germination methods, not only differentiated by the method of moisturizing, temperature, and access to light, but also by the substrate [6,10,11]. Studies on the processes occurring in germinating seeds also explore the importance of light as an intermediary in the regulation of enzyme activity [12]. Seed viability can be expressed as the ability to germinate, which leads to the formation of a plant capable of reproduction. Sufficiently long viability affects the vigour of seeds, which expresses their ability to produce healthy and well-growing seedlings and plants. The cultivation of seeds both in terms of their germination efficiency and the content of bioactive compounds can be stimulated in various ways. An interesting factor is, for example, the usage of the pulsed electric field (PEF) on seeds before cultivation. The usage of the pulsed electric field on seeds is a phenomenon described in various ways in the literature. It has been shown that the effect of the PEF on seeds before germination has a positive effect on germination efficiency and seed growth rate [13,14]. Research was also conducted on the use of PEF [15–20], magnetic field [21] and UV light [22] in food processing and preservation. In biological material that has been subjected to an electric field, as a result of the impact of electrical impulses on cell membranes, a significant increase in their conductivity is observed, mainly through the formation of free spaces on the surface, the so-called pores. Their presence allows the free flow of various components through the cell membrane. Thanks to this phenomenon, it becomes possible to transfer ions, molecules and even more complex compounds (i.e., drugs, nucleic acids, monoclonal antibodies, oligonucleotides or plasmids) into the cell [23].

It is believed that the PEF can potentially be used to control and optimize the process of sprout growth and modify its composition, in particular nutritional values and bioactive ingredients [24]. PEF can also effectively stimulate germ growth and positively affect metabolism and nutrient content [24,25].

The effect of PEF on germination depends on the type of plant and the strength of the used field, while on some species such as marigold tomato or radish no significant effect was observed. In the case of lentil, a 50% increase in germination rate was observed. The effect of changes in germination rate induced by PEF is probably associated with changes in the metabolism of amino acids occurring in seeds [25] while PEF induces electroporation causing increased membrane permeability. Electroporation is a reversible process, however, when too high values of the PEF process parameters are used, irreversible changes in the structure of the cell membrane can occur [26]. Depending on the duration and the number of pulses and the strength of the electric field, the cell membrane may even be destroyed (i.e., irreversible electroporation) [25].

Red clover *Trifolium pratense* L. is an interesting plant. Throughout the world, it is most often known as a feed plant because it is used for fodder. There are many publications on bioactive compounds and their variability resulting from the cultivation of a mature plant. Many pharmaceutical

preparations and dietary supplements [27,28] are also obtained on the basis of this plant. There is little research on carotenoids in red clover, and no such data were found in the available literature. Research and modelling of carotenoids content in red clover sprouts are innovative.

It should be noted that it has been confirmed that red clover sprouts are a rich source of bioactive compounds, including isoflavones, compounds with similar effects to oestrogen and can supplement the daily diet [6].

Another factor more and more often considered in the scientific literature, regarding the plants' cultivation, including sprouts, is the effect of light of different wavelengths on growth factors, as well as the content of biologically active compounds, including carotenoids [29]. Light, its intensity and wavelength have a significant impact on germination and plant development. Analysing the reports of various authors, the impact of light of varying wavelengths on sprouts of plants of different species is different. There is no information on the effect of light of varying wavelengths on the content of carotenoids in red clover sprouts. Therefore, the purpose of this work is to attempt to model the content of carotenoids in red clover sprouts using the effects of the PEF on their cultivation and the light of different wavelengths during their growth.

## 2. Materials and Methods

The experiment was carried out for seven days under different growing conditions. In the experiment, different wavelengths of light and pulsed electric fields (PEF) were used. The light-emitting diodes (LED) were used as the source of light. In the conducted research the light of wavelengths, ranging from 340–780 nm, was applied to the tested material. The strength of the pulsed electric field was chosen from the ranges 1 kV/cm, 2.5 kV/cm and 5 kV/cm. During the experiment, favourable conditions for seed development were created. The red clover seeds were put in a phytotron chamber where the light of different wavelengths was applied. Another group of seeds was treated with PEF before sowing and then they were cultivated in white light conditions. Cultivation of seeds was carried out to obtain the edible sprouts. The collection time was determined based on the quality features and content of bioactive compounds in red clover sprouts, including ascorbic acid and flavonoids (phytoestrogens). These compounds are present in the largest quantities between the 6th and 8th day of cultivation [6,11]. Thus, the germination in this experiment took place after 7 days of cultivation (after 168 h).

In the experiment, the sprouts were grown without soil and without supporting substrates containing mineral substances. Therefore, further development of the plant would be inhibited as a result of depletion of spare substances stored in seeds. Since whole sprouts of red clover are intended for consumption, the division into cotyledon and hypocotyl was not included in the study. It was found that on the day of harvesting red clover sprouts were firm, had green leaves with a cucumber-pea smell and were properly shaped, but depending on the cultivation conditions, the obtained weight (biomass) varied.

### 2.1. Pulsed Electric Field

A laboratory stand was built to conduct the experiment. It consisted of a high-voltage pulse generator with the voltage output in the range of 0 to 30 kV. The generated signal was the rectangular shape. The control system was used to obtain the proper value of the electric field strength. It allowed setting the number of pulses and the time interval between them. The process was conducted in the chamber in which the discharge occurred. There were two flat electrodes, between which a cylindrical, Teflon cell with seeds was placed.

The following parameters of the electroporation process were selected: pulse repeatability 10 s; the number of impulses 20; the electric field strengths 1 kV/cm, 2.5 kV/cm and 5 kV/cm. Higher field strength values were not used due to the destruction of the tested material. During the process, the constant value of temperature 20 °C was ensured. The temperature was measured with a thermocouple. The PEF was applied to red clover seeds before germination.

### 2.2. Raw Material

The seeds of red clover (*Trifolium pratense L.*), Rosette variety, suitable for germination supplied by FN Granum (Wodzierady, Poland) were chosen as the tested material. The plants were divided into 49 experimental series. Each series consisted of 5 g seeds. In each series there were three containers.

### 2.3. Sprouts Cultivation

Red clover (*Trifolium pratense* L.) seeds were grown using modified conditions. Seeds germination was carried out in the climatic chamber with a phytotron system (modified KBWF 720 Binder, Tuttlingen, Germany). Cultivation of the sprouts was carried out for 7 days at relative humidity 80% and 20 ± 1 °C. The LEDs which emitted the light with different wavelengths were used as the light sources in the process of sprouts growing. The different light wavelengths were used during cultivation: white light (380–780 nm), UVA (340 nm), blue (440 nm), red (630 nm). A photosynthetic photon flux density (PPFD) of 150 ± 5 $\mu$mol m$^{-2}$ s$^{-1}$ was maintained.

Since it is not possible to use sunlight in the phytotron chamber, white light (cool white) was used for all experimental samples to maintain constant growing conditions. Such lighting was also used by other authors in the cultivation of plants, including sprouts [30]. It would be possible to use sunlight in the experiment, but the sunlight would be variable in time and the environmental conditions would vary (e.g., in sprout machine).

In addition, the pulsed electric field (PEF) with three values of strength, 1, 2.5 and 5 kV/cm, respectively were applied on three series of sprouts before sowing. Seeds after PEF treatment were stored for 14 days at room temperature (T = 20 ° C) and then grown.

Sprouts treated with PEF were grown under white light (380–780 nm).

During cultivation, lighting was used repeatedly for 12 h and after that period the plants were stored 12 h in the dark. Sprouts were cultivated in portions of 5 g in containers (polypropylene) with a perforated bottom set on a tray. Sprouting was conducted in triplicate for each treatment (3 containers for each experiment). On the first day, the seeds were soaked by the addition of 15 mL of water, and in the subsequent days, the examined sprouts were irrigated to maintain high moisture in the culture. During cultivation, sprout weight growth was analysed every 24 h. Red clover sprouts were weighed immediately after harvesting on an analytical balance (RADWAG, PS 06.R2, Radom, Poland). The obtained germs of the sprouts are the average of three samples of 5 g seeds each. From the time the seeds were sown, the samples were weighed every 24 h.

The whole sprouts were examined. Sprouts were harvested manually every 24 h from the day of sowing. Analyses from the average sample from each container (3 containers for each treatment) were performed in one test samples taken from each container. After each harvest, the weight of the harvested sprouts was determined. The content of carotenoids was determined after 7 days of red clover sprouts growing.

These samples after harvesting were frozen immediately in liquid nitrogen and stored at −80 °C until further analysis.

### 2.4. Germination Energy

For each cultivation variant, four hundred seeds from the International Seed Testing Association (ISTA) (red clover and red clover after PEF treatment) were collected, grown in separate moulds made of polipropylen (PP) plastic, dedicated to conducting the germination process. During the study, the percentage of healthy, correctly germinated seeds was determined. During the cultivation, seed viability indexes were determined for energy of red clover seeds.

Germination energy (GE) was determined as the percentage of seeds that germinated during the first 4 days (96 h) [31].

## 2.5. Sample Preparation and Determination of Carotenoids

The samples of sprouts were placed in liquid nitrogen using an analytical mill (A 11 basic, IKA Works GmbH & Co. KG, Staufen, Germany)

The method proposed by Kimura and Rodriguez-Amaya [32] with some modifications was used for the extraction of carotenoids. First, 1 g of grounded and homogenized sprouts was placed in a plastic tube. Extraction was performed twice, first with 5 mL of acetone and next with 5 mL of ethanol (100%). Each extraction step was performed by vortex mixing for 2 min followed by centrifugation at 15,000 RCF (relative centrifugal force) for 10 min at 4 °C (MPW-360R, MPW Med. Instruments, Warsaw, Poland). The supernatants were evaporated to dryness under a stream of $N_2$ at 35 °C. The residue was redissolved in the mobile phase, centrifuged and diluted with the mobile phase to a suitable concentration.

The carotenoid extracts (injection volume 20 μL) were analysed on an HPLC system (Knauer Wissenschaftliche Geräte GmbH, Berlin, Germany) comprising of degasser of mobile phase Manager 5000, Pump 1000, autosampler model 3935 (maintained at 6 °C). A Gemini column maintained at 40 °C was used (5u C18110A, 150 × 4,60 mm 5 μm, Phenomenex, Torrance, CA, USA). The HPLC system was coupled with a photodiode array detector (PDA) model 2800. Quantification was performed at a detection wavelength of 445 nm. A gradient system (from A:B 75:25 to 0:100 in 13 min, maintaining this proportion until the end of the turn) was applied with methanol:water (90:10 *v/v*) as eluent A and acetonitrile:2-propanol (63:37 *v/v*) as eluent B. The flow rate was 1 mL/min. The carotenoids were identified based on the retention time of the standards and quantified with the help of peak areas against the standard calibration curves. Standards of carotenoids were purchased from Sigma-Aldrich (Saint Louis, MO, USA).

## 2.6. Statistical Analysis

Statistical analysis was based on the determination of the average values of three measurements and their standard deviation. The data were analysed using a one-way analysis of variance (ANOVA) with the Tukey's range test at the significance level $p < 0.05$. All data were tested for normality using the Shapiro–Wilk test. To test for homogeneity of variance, Levene's test was used. The calculations were performed using the software STATISTICA for Windows (version 10, Statsoft, Krakow, Poland).

## 3. Results and Discussion

In the experiment, a research hypothesis was put forward that it is possible to modify the content of carotenoids in red clover sprouts (*Trifolium pratense* L) by applying PEF to seeds before sowing and using light of varying wavelengths during cultivation. Until now, these factors have not been combined in such a way. Thus, this is an innovative approach to this issue. The highest sprout mass was obtained after applying red light to the seeds (38.9 g) and in the case of seeds cultivated in white light after PEF (5 kV/cm) pre-treatment (35.5 g). The lowest final mass was obtained in the case of UVA irradiation.

The seeds cultivation in white light after PEF pre-treatment with the field strengths of 1, 2.5 and 5 kV/cm increased the yield by 0.38%, 3.37% and 6.12%. Table 1 presents the final mass of the 100 g samples after 7 days of cultivation (on the day of harvest) in the case of different conditions: in UVA, blue, red and white light, without and after PEF pre-treatment with the field strengths of 1, 2.5 and 5 kV/cm. (Table 1). Statistically significant differences between the experimental series are presented in Table 2.

**Table 1.** Germination energy given per 100 pieces of seeds and the final mass of the samples after their cultivation in the different growth conditions, recalculated to the initial weight of 100 g sprouts. Values are presented as mean ± SD.

| Light | Germination Energy (%) * ($n = 4$) | Final Mass (g) ** ($n = 3$) |
|---|---|---|
| UVA (340 nm) | 89.3 ± 2.9 | 600.8 ± 36.6 |
| blue (440 nm) | 88.5 ± 2.1 | 629.0 ± 30.2 |
| red (630 nm) | 88.5 ± 2.3 | 778.6 ± 34.3 |
| white (380–780 nm) | 89.0 ± 2.3 | 669.5 ± 28.1 |
| **White Light and Pulsed Electric Field Strength** | **Germination Energy (%) *** ($n = 4$) | **Final Mass (g) *** ($n = 3$) |
| 1.0 kV/cm | 88.8 ± 2.3 | 672.0 ± 21.7 |
| 2.5 kV/cm | 90.8 ± 2.9 | 692.0 ± 20.8 |
| 5.0 kV/cm | 94.0 ± 3.0 | 710.4 ± 30.5 |

* The performed analysis of variance does not allow for the rejection of the hypothesis of equality of means (F = 1.82, $p = 0.143$). Kruskal–Wallis test was conducted to examine the differences on germination energy according to the treatment taken. No significant differences (Chi-squared = 10.91, $p = 0.09$) were found among all the seven growing conditions. ** Statistically significant differences are presented in the Table 2.

**Table 2.** *p*-Values of Tukey's post hoc tests after one-way ANOVA for final mass of sprouts with treatments as fixed factors (F = 11.50, $p < 0.05$). Statistically significant differences are bolded.

| Treatment | UVA | Blue | Red | White | PEF 1 | PEF 2.5 |
|---|---|---|---|---|---|---|
| UVA | - | - | - | - | - | - |
| blue | 0.8931 | - | - | - | - | - |
| red | **0.0002** | **0.0005** | - | - | - | - |
| white | 0.1299 | 0.6335 | **0.0066** | - | - | - |
| PEF 1 | 0.1094 | 0.5735 | **0.0079** | 1.0000 | - | - |
| PEF 2.5 | **0.0261** | 0.1965 | **0.0342** | 0.9637 | 0.9796 | - |
| PEF 5 | **0.0063** | 0.0522 | 0.1345 | 0.6254 | 0.6850 | 0.9827 |

Other authors also show a differentiated effect on the growth and yield of different plants depending on the wavelength of the used light. The results of the research indicate that the usage of red LED light in the cultivation of seeds and plants can cause an increase in biomass [33,34]. The red light is the most effective photosynthetically active radiation [35] referring to absorption. Research shows that blue light can also be used as a factor to increase biomass [36].

In the case of crops of different varieties of lettuce sprouts, it was found that the fresh weight increased in the case of using red or blue light in the growing process compared to crops under white light. The sprouts were irradiated for 12 h and left in the dark for 12 h. The best effect of the growth of germ mass was obtained when the sprouts were exposed to 3 h of blue light, and in one case it was observed that it increased by 23% [8]. On the other hand, in other studies [25] where wheat seeds before cultivation were treated with 50 pulses of 6 kV/cm PEF, a significant increase in fresh weight was noted, which is positively correlated with PEF energy.

A statistically significant effect of PEF (used at the initial stage of seed preparation for cultivation) on the mass of obtained red clover sprouts was observed. The exposure of seeds to the pulsed electric field can activate processes at an early stage of photomorphogenesis and thus positively affect its development. That line of research requires continuing experiments to confirm this observation. It was not the primary purpose of the presented studies.

In the case of wheatgrass (*Triticum aestivum* L.) the usage of 0.5 kV/cm PEF did not affect the growth of sprouts compared to the control ones. Increasing the PEF strength to 1.4 kV/cm had a positive effect on sprout growth, however, the 2 kV/cm PEF treatment of the tested seeds had an adverse effect and a lower growth was obtained than compared to the control [24].

Figure 1 shows the mass of sprouts depending on the day of cultivation. The initial mass of each sample was 5 g. The obtained mass in the harvest day depended on the pre-treatment and cultivation conditions.

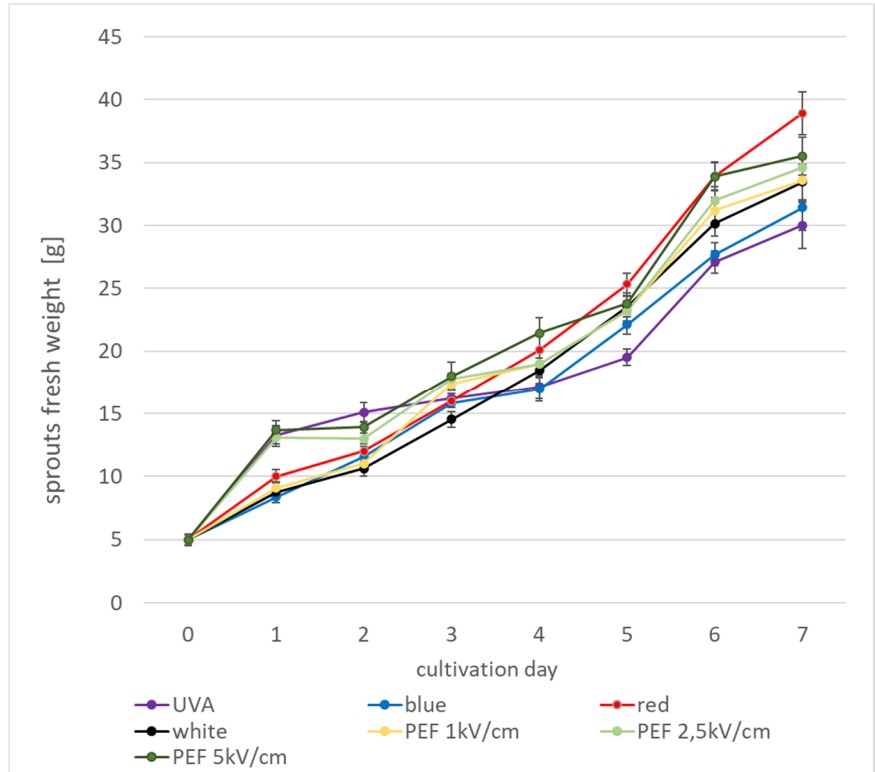

**Figure 1.** Mass of red clover sprouts during 7 days of sprout growth depending on irradiation and pulsed electric field (PEF) treatment prior to sowing. Values are presented as mean ± SD (*n* = 3).

The germination energy of red clover seeds (Table 1) in the case of PEF 5 kV/cm (in the process of seed pre-treatment before sowing) was the highest but did not differ statistically in comparison to other treatments. Moreover, all analysed groups had no significant differences in germination energy. The use of monochromatic LED sources and white light with the addition of 1 and 2.5 kV/cm PEF did not affect germination energy compared to white light crops (no statistically significant differences were found).

A statistically significant increase in germination energy was found after 4 days using a field of energy 240 J (12 kV) for parsley and 960 J (12 kV, the number of pulses was changed) for parsley, winter wheat, winter barley, lettuce, tomato and garden rocket [37].

In other studies, germination energy measured after 24 h was positively correlated with the increase in intensity (energy of pulse electric field) and in the case of 6 kV/cm PEF with 50 pulses it was 92% compared with 84% for non-treated sprouts [25].

For *Haloxsylon ammodendron* seeds, it has been observed that germination energy increases from 72.9% to 90.3% and 98.0% in the cases of using the strengths of 10 and 20 kV/cm, respectively [38].

It was observed that red clover sprouts after 7 days of cultivation contain β-carotene, lutein and zeaxanthin. However, their content varies depending on the factors used (PEF, different lengths of light). In this experiment, monochromatic light sources were used in the process of red clover sprouts growing. Moreover, the chosen wavelengths had a documented effect on photosynthesis and photomorphogenesis [36,39–41].

Green light was not chosen to process the experiment because it was documented in studies that a green monochromatic light source did not affect sprout development. Only slight enrichment of the spectrum with other light wavelengths was contemplated to improve photomorphogenesis [39,42].

In this study, it was observed that the content of carotenoids depends on the conditions of the used light (Figure 2). The dominant carotenoid in germinating red clover seeds is lutein, the content of which varies from 743 mg/kg in sprouts grown in red light, to 862 mg/kg in sprouts grown in UVA, and 888 mg/kg in sprouts grown in blue light. Lutein is a carotenoid with widely documented health-promoting properties. Obtaining new sources of lutein is desirable. At the same time, it was observed that the use of UVA or blue light during cultivation had a positive effect on the formation of other carotenoids, including β-carotene and zeaxanthin. UVA and blue light are the most preferred for obtaining carotenoid dyes. Under these cultivation conditions, the highest content of the total value of tested carotenoids (β-carotene, lutein and zeaxanthin) in red clover sprouts was obtained, respectively 1750 mg/kg (kg of dry weight) in the case of UVA irradiation and 1892 mg/kg using blue irradiation. Comparing these results to sprouts grown in white light, the increase in the content of these dyes in sprouts irradiated with blue light is statistically significant and equal to 42% in β-carotene, 19% in lutein and 14% in zeaxanthin (Figure 2, Tables 3–5). The content of lutein in sprouts from crops cultivated under red light was slightly lower than under white light, but the difference is not statistically significant (Table 4).

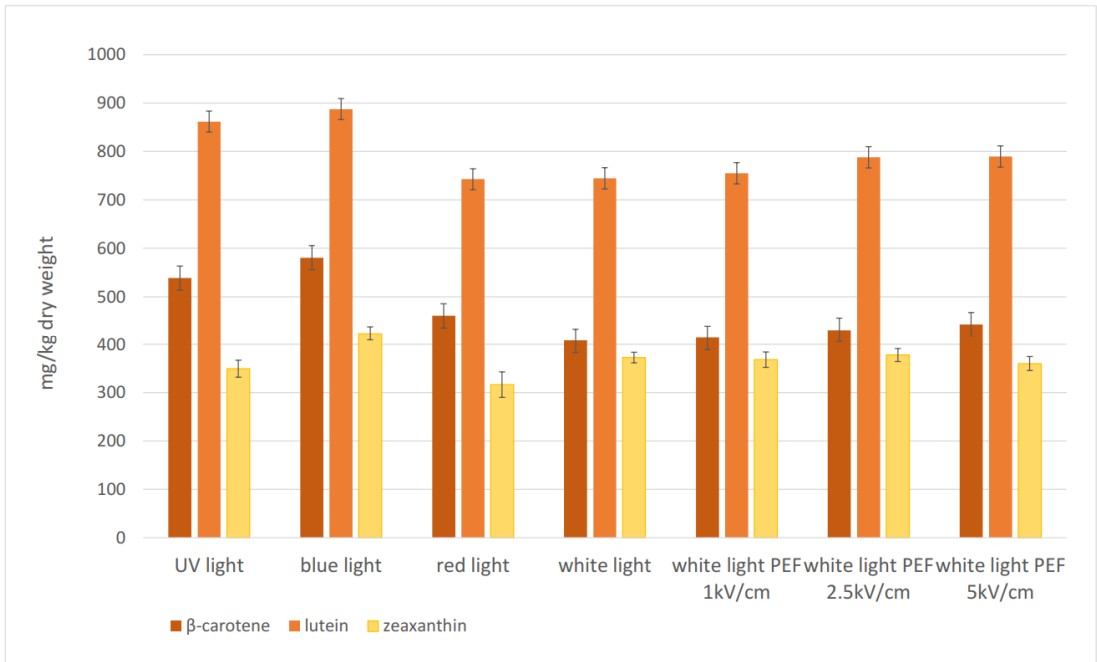

**Figure 2.** Influence of different light and PEF pre-treatment on carotenoids. The height of each bar and the error bars indicate the means and standard errors, respectively, from three independent measurements.

**Table 3.** *p*-Values of Tukey's post hoc tests after one-way ANOVA for β-carotene with treatments as fixed factors (F = 30.84, *p* < 0.05). Statistically significant differences are bolded.

| Treatment | UVA | Blue | Red | White | PEF 1 | PEF 2.5 |
|-----------|-----|------|-----|-------|-------|---------|
| UVA | - | - | - | - | - | - |
| blue | 0.2244 | - | - | - | - | - |
| red | **0.0056** | **0.0002** | - | - | - | - |
| white | **0.0002** | **0.0002** | 0.0866 | - | - | - |
| PEF 1 | **0.0002** | **0.0002** | 0.1591 | 0.9997 | - | - |
| PEF 2.5 | **0.0004** | **0.0002** | 0.6018 | 0.8137 | 0.9470 | - |
| PEF 5 | **0.0010** | **0.0002** | 0.9321 | 0.4290 | 0.6354 | 0.9911 |

**Table 4.** *p*-Values of Tukey's post hoc tests after one-way ANOVA for lutein with treatments as fixed factors (F = 21.10, *p* < 0.05). Statistically significant differences are bolded.

| Treatment | UVA | Blue | Red | White | PEF 1 | PEF 2.5 |
|-----------|--------|--------|--------|--------|--------|---------|
| UVA | - | - | - | - | - | - |
| blue | 0.7675 | - | - | - | - | - |
| red | **0.0003** | **0.0002** | - | - | - | - |
| white | **0.0003** | **0.0002** | 1.0000 | - | - | - |
| PEF 1 | **0.0006** | **0.0002** | 0.9915 | 0.9966 | - | - |
| PEF 2.5 | **0.0133** | **0.0011** | 0.2138 | 0.2511 | 0.5309 | - |
| PEF 5 | **0.0156** | **0.0013** | 0.1859 | 0.2194 | 0.4801 | 1.0000 |

**Table 5.** *p*-Values of Tukey's post hoc tests after one-way ANOVA for zeaxanthin with treatments as fixed factors (F= 11.07, *p* < 0.05). Statistically significant differences are bolded.

| Treatment | UVA | Blue | Red | White | PEF 1 | PEF 2.5 |
|-----------|--------|--------|--------|--------|--------|---------|
| UVA | - | - | - | - | - | - |
| blue | **0.0015** | - | - | - | - | - |
| red | 0.2638 | **0.0002** | - | - | - | - |
| white | 0.6442 | **0.0287** | **0.0151** | - | - | - |
| PEF 1 | 0.8147 | **0.0163** | **0.0266** | 0.9999 | - | - |
| PEF 2.5 | 0.4240 | 0.0575 | **0.0075** | 0.9996 | 0.9903 | - |
| PEF 5 | 0.9842 | **0.0057** | 0.0755 | 0.9669 | 0.9963 | 0.8470 |

Carotenoid content was studied in mature red clover plants for fodder grown in the field. Research shows that the main carotenoid is lutein, its content was 136 mg/kg DM (dry matter), and the total β-carotene content was 29 mg/kg DM [43]. In other studies, the composition of the carotenoids in red clover is different, and the content of lutein, β-carotene and zeaxanthin is much higher, respectively 237.7, 100.3 and 91.7 mg/kg DM. It should be noted that in recent studies, clover after harvesting has been subjected to a consolidation process [44].

Other authors report that the content of β-carotene in red clover is about 200 mg/kg DM, and the content of this carotenoid in *Trifolium repens* varies from 300 to 730 mg/kg DM depending on the harvesting place [45].

In the experiment where the subject was Alsike clover, also belonging to the genus *Trifolium*, the content of lutein and β-carotene was much higher and equalled 208.9–243 mg/kg and 35.4–123.4 mg/kg fresh weight, respectively [46].

The effect of various types of LED lighting on the content of carotenoids in alfalfa sprouts (also belonging to *Fabaceae* family) was investigated [47]. The combination of the red and blue LEDs used during cultivation increased lutein content from 82.6 to 108.2, and β-carotene from 26.6 to 44.6 mg/kg fresh mass of sprouts [47]. However, the research did not cover the whole sprout, only cotyledon. Increasing the intensity of blue light during the cultivation of beetroot (*Beta vulgaris* L.) and parsley (*Petroselinum crispum* Mill.) caused an increase in the content of carotenoids as compared to irradiation with red light. It has been shown that increasing the proportion of blue light intensity in these microgreens results in an increase in lutein content from 103.8 to 118.5 mg/kg of fresh beet mass, from 122.9 to 190.7 mg/kg of fresh parsley, and β-carotene, respectively from 0.09 to 0.87 and 0.54 to 0.86 mg/kg, and zeaxanthin from 1.39 to 3.20 and up to 0.84–14.4 mg/kg, respectively [35].

Lefsrud in his research observed an increase in the content of lutein and β-carotene in kale when illuminated with both blue and red light. The maximum accumulation of lutein and β-carotene in fresh kale mass was found at 640 nm and 440 nm, respectively, calculated on a fresh mass basis. However, when converted to dry mass, the maximum lutein content in kale also occurred at 440 nm. [1].

There are no reports in the literature on the content of carotenoids in red clover sprouts, as well as on the impact of PEFs on the content of carotenoids. Therefore, our research can only be compared with the unique research carried out by Ahmed et al. [25] who studied the effect of PEF on the content

of plant dyes in wheat germ. A 35% increase in the content of carotenoids was observed in the PEF treatment with strength of 6 kV/cm and 50 impulses compared to untreated sprouts [25].

In our experiment, no such significant effect of PEF treatment of red clover seeds on the content of carotenoids was found.

Modelling the content of carotenoids in sprouts which are grown in different lighting conditions, especially in ultraviolet radiation with wavelengths up to 400 nm and blue light, forces the plant to adapt to the changes to avoid their harmful effects. Many compounds (e.g., flavonoids as well as carotenoids) can fulfil the role of protection against the harmful effects of UV light [6,48]. Carotenoids are the auxiliary photoreceptors of chlorophyll and absorb light mainly in the blue region. Cryptochromes are the receptors of blue light in the 390–480 nm range. They stimulate leaf expansion. In the blue and UVA spectrum, phototropins, responsible for plant phototropism, bending of shoots towards the light, opening of stomata and leaf expansion are also photoreceptors in the blue and UVA spectrum [49,50].

The increase of carotenoids content is plants' response to stress is associated with high irradiance. High exposure does not always lead to the growth of carotenoids, it can sometimes cause photodegradation of pigment particles [1,51,52]

In the presented experiments, we observed that monochromatic blue light increases the content of carotenoids. The positive effect of blue light on the content of these dyes was also observed by other authors [52,53] Opposite observations were made by Tuan et al. [29]. Research on tartary buckwheat sprouts has shown that blue light causes a decrease in the content of carotenoids compared to white light. However, it should be noticed that in the recalled work, the full spectrum range is not given in the description of the characteristics of the white light source (only 380 nm is mentioned (i.e., UV wavelength)).

The effect of pulsed electric field on seeds (before cultivation) had an impact on the increase of carotenoid content, including β-carotene, lutein and zeaxanthin. It was observed that the application of 5 kV/cm had the most favourable effect on the increase in the content of carotenoids in red clover sprouts compared with the reference test which grew seeds in white light. An increase in the content of β-carotene and lutein in red clover sprouts was obtained in comparison to the test with white light and without PEF pre-treatment by 8.5% and 6%, respectively, and a decrease in the content of zeaxanthin by 3.3% was also found. Therefore, PEF pre-treatment may increase mainly the content of β-carotene. At the same time, it should not be assumed that the use of a higher value of electric field strength (above 5 kV/cm) will work more favourably. The use of higher field strength can damage or destroy the seed epidermis and damage its internal structure, which means that the seed will not be able to germinate. The results of this experiment indicate that the PEF pre-treatment on seeds before their cultivation can be one of the factors that can model the carotenoids content in germinated seeds. The obtained results should be considered as an introduction to further research.

Photosynthetically active radiation includes a wide range of light wavelengths that do not participate in the process of photosynthesis but can stimulate the vegetative growth of plants and modify the chemical composition of the leaves [54].

## 4. Conclusions

The dominant carotenoid in germinating red clover seeds is lutein, whose content varies from 743 mg/kg in sprouts grown in red light, 862 mg/kg in sprouts grown in UVA, to 888 mg/kg in sprouts grown in blue light. UVA and blue light are the most preferred for obtaining carotenoid dyes. The highest content of the total value of tested carotenoids (β-carotene, lutein and zeaxanthin) in red clover sprouts was obtained, respectively at 1750 mg/kg (kg of dry weight) in the case of UVA irradiation and 1892 mg/kg using blue irradiation.

Modelling the content of carotenoids in red clover sprouts is possible because by using UVA and blue light (440 nm) during seed cultivation, a significant increase in β-carotene, lutein and zeaxanthin is obtained. An interesting result of the presented research is also the use of PEF pre-treatment on the seeds before the cultivation process. It is a factor that, although to a lesser extent than blue light and

UVA, also causes an increase of the carotenoids in sprouts. It is possible that PEF pre-treatment of seeds will also have a beneficial effect on other seeds intended for germination. It requires further testing and confirmation. At the same time, the use of red light resulted in a greater mass of red clover sprouts, because red light has a beneficial effect on photomorphogenesis.

Consumption of 100 g fresh red clover sprouts which were grown for 7 days in UVA, blue, red or white light provided 14.0, 15.1, 12.1 and 12.2 mg total sum of tested carotenoids, respectively. Although the PEF treatment does not significantly affect the carotenoids' content, in the case of PEF strength 5 kV/cm the intake of carotenoids increased by 4% compared to cultivation only in white light and equals 12.7 mg/100 g.

It is worth noting that red clover sprouts may also be a new source of carotenoids including lutein and zeaxanthin. The recommended dietary allowances (RDA) of lutein and zeaxanthin in the human diet have not been established. For reducing the risk of AMD, the efficacious intake level for lutein may be ~ 6 mg per day [55]. Therefore, the consumption of approx. 40 g of red clover sprouts may cover the daily requirement.

Cultivation of seeds for sprouts in closed conditions (in a phytotron chamber) with properly selected blue or UVA lighting can stimulate an increase in the content of bioactive compounds, including carotenoids, and at the same time ensure the proper development of leaves. Changing the spectral composition of radiation during cultivation can be used to produce plants intended for several days of germination and consumption. It is possible to obtain a significantly improved product in terms of nutritional value in a short time using the tested conditions.

**Author Contributions:** Conceptualization, I.G.-C. and E.K.; methodology, I.G.-C., E.K. and P.K.; validation, I.G.-C., E.K., A.C., P.K. and T.D.; formal analysis, I.G.-C. and E.K.; investigation, I.G.-C., E.K., A.C., P.K. and T.D.; resources, I.G.-C. and P.K. and T.D.; data curation, I.G.-C., P.K. and T.D.; writing—original draft preparation, I.G.-C., E.K. and A.C.; writing—review and editing, I.G.-C., E.K. and A.C.; visualization, A.C.; supervision, I.G.-C.; project administration, I.G.-C. and E.K.; funding acquisition, I.G.-C.; P.K. and T.D. All authors have read and agreed to the published version of the manuscript.

**Funding:** This research received no external funding.

**Conflicts of Interest:** The authors declare no conflicts of interest.

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
