# Peer review of "Modelling of Carotenoids Content in Red Clover Sprouts Using Light of Different Wavelength and Pulsed Electric Field"

_applsci, doi:10.3390/app10124143_

Round 1

Reviewer 1 Report

In chapter Materials and methods I suggest specifying the elution gradient program used in HPLC analysis.

In the Table 1. the authors should provide statistical probability and the significant differences between the treatments since in the text contains the comparisons of the treatment sprouts.

In Figure 2. the authors compared lutein, zeaxanthin and β-caroten at each treatment using one-way ANOVA test. That kind of statistical analysis makes no sense. I suggest comparing each carotenoid content across the treatments using the one-way ANOVA, if data of each caroteniud are normal and the variances are equal. The text of the manuscript should be adopted according to the statistical analysis described above. 

Author Response

Dear Reviewer,

We would like to thank you very much for your time you spent for correcting our paper and suggestions. They have helped us to improve the paper.

Below we have written answers for your comments. All of them are placed directly under the comments and they are highlighted in the text :

In chapter Materials and methods I suggest specifying the elution gradient program used in HPLC analysis.

The additional information about gradient system has been placed in the text in the Materials and Methods section:

“A gradient system (from A:B 75:25 to 0:100 in 13 min, maintaining this proportion until the end of the turn) was applied”

In the Table 1. the authors should provide statistical probability and the significant differences between the treatments since in the text contains the comparisons of the treatment sprouts.

The additional information has been placed in the section Statistical analysis:

“All data were tested for normality using the Shapiro-Wilk test. To test for homogeneity of variance, Levene’s test was used.”

Statistically significant differences are supplemented in Table 1. Now the Table is as below:

Table 1. Germination energy of given per 100 pieces of seeds and the final mass of the samples after their cultivation in the different growth conditions, recalculated to the initial weight of 100g sprouts. Values are presented as: mean ± SD.

light

germination energy [%]*

(n = 4)

final mass [g]**

(n = 3)

UVA (340 nm)

89.3 ± 2.9

600.8 ± 36.6

blue (440 nm)

88.5 ± 2.1

629.0 ± 30.2

red (630 nm)

88.5 ± 2.3

778.6 ± 34.3

white (380-780 nm)

89.0 ± 2.3

669.5 ± 28.1

white light and pulsed electric field pretreatment

field strength

1.0 kV/cm

88.8 ± 2.3

672.0 ± 21.7

2.5 kV/cm

90.8 ± 2.9

692.0 ± 20.8

5.0 kV/cm

94.0 ± 3.0

710.4 ± 30.5

* The performed analysis of variance does not allow for the rejection of the hypothesis of equality of means (F=1.82, p=0.143). Kruskal-Wallis test was conducted to examine the differences on germination energy according to the treatment taken. No significant differences (Chi square =10.91, p = 0.09) were found among all the seven growing conditions.

**  Statistically significant differences are presented in the table 2.

Table 2. P values of Tukey post hoc tests after one-way ANOVA for final mass of sprouts with treatments as fixed factors, F=11,50, significance level p< 0.05. Statistically significant differences are bolded

treatment

UVA

blue

red

white

PEF 1

PEF 2.5

UVA

blue

0.8931

red

0.0002

0.0005

white

0.1299

0.6335

0.0066

PEF 1

0.1094

0.5735

0.0079

1.0000

PEF 2.5

0.0261

0.1965

0.0342

0.9637

0.9796

PEF 5

0.0063

0.0522

0.1345

0.6254

0.6850

0.9827

In Figure 2. the authors compared lutein, zeaxanthin and β-caroten at each treatment using one-way ANOVA test. That kind of statistical analysis makes no sense. I suggest comparing each carotenoid content across the treatments using the one-way ANOVA, if data of each carotenoid are normal and the variances are equal. The text of the manuscript should be adopted according to the statistical analysis described above.

To improve the manuscript according to reviewer’s suggestion, Figure 2 was changed and the proper values have been written in tables. In Tables 2-5 we have presented the P values of Tukey post hoc tests and outlined the statistically significant differences for each carotenoid separately. A similar method of presenting statistically significant differences was used for the final germ mass in Table 1. For germination energy it was necessary to use the nonparametric Kruskal-Wallis test.

Figure 2. Influence of different light  and PEF pre-treatment on carotenoids. The height of each bar and the error bars indicate the means and standard errors, respectively, from three independent measurements. Statistically significant differences in the average content of each (individual) carotenoid depending on the growing conditions are presented in the tables 3-5.

Table 3. P values of Tukey post hoc tests after one-way ANOVA for β-carotene with treatments as fixed factors, F=30.84, significance level p< 0.05. Statistically significant differences are bolded

treatment

UVA

blue

red

white

PEF 1

PEF 2.5

UVA

blue

0.2244

red

0.0056

0.0002

white

0.0002

0.0002

0.0866

PEF 1

0.0002

0.0002

0.1591

0.9997

PEF 2.5

0.0004

0.0002

0.6018

0.8137

0.9470

PEF 5

0.0010

0.0002

0.9321

0.4290

0.6354

0.9911

Table 4. P values of Tukey post hoc tests after one-way ANOVA for lutein with treatments as fixed factors, F=21.10, significance level p< 0.05. Statistically significant differences are bolded.

treatment

UVA

blue

red

white

PEF 1

PEF 2.5

UVA

blue

0.7675

red

0.0003

0.0002

white

0.0003

0.0002

1.0000

PEF 1

0.0006

0.0002

0.9915

0.9966

PEF 2.5

0.0133

0.0011

0.2138

0.2511

0.5309

PEF 5

0.0156

0.0013

0.1859

0.2194

0.4801

1.0000

Table 5. P values of Tukey post hoc tests after one-way ANOVA for zeaxanthin with treatments as fixed factors, F=11,07, significance level p< 0.05). Statistically significant differences are bolded.

treatment

UVA

blue

red

white

PEF 1

PEF 2.5

UVA

blue

0.0015

red

0.2638

0.0002

white

0.6442

0.0287

0.0151

PEF 1

0.8147

0.0163

0.0266

0.9999

PEF 2.5

0.4240

0.0575

0.0075

0.9996

0.9903

PEF 5

0.9842

0.0057

0.0755

0.9669

0.9963

0.8470

 All numbers of tables and references have been corrected.

Attached please find the improved and upgraded text of our article.

Reviewer 2 Report

The objective of the study is to evaluate the effect of the light and pulsed electric field on the carotenoid concentration in red clover spouts. Several light wavelengths were tested. The study is interesting and original. The article is well written and the results of the study are well presented. However the authors should improve the quality of the manuscript in order to be accepted for publication regarding clarity of the presented results and data interpretation. More specifically, the Discussion section should be expanded and the results should be compared quantitatively with similar studies from the literature. Additionally,  the statistical analysis is not evident and the authors do not present any statistical data regarding the significant differences between the obtained results. The abstract should be rewritten and more explicitly present the main scope and results of the study. E.g. the sentence in lines 14-15 should be completed (carotenoids content in what?)

Author Response

Dear Reviewer,

We would like to thank you very much for your time you spent for correcting our paper and suggestions. They have helped us to improve the paper.

Below we have written answers for your comments. All of them are placed directly under the comments and they are highlighted in the text :

The objective of the study is to evaluate the effect of the light and pulsed electric field on the carotenoid concentration in red clover spouts. Several light wavelengths were tested. The study is interesting and original. The article is well written and the results of the study are well presented. However the authors should improve the quality of the manuscript in order to be accepted for publication regarding clarity of the presented results and data interpretation. More specifically, the Discussion section should be expanded and the results should be compared quantitatively with similar studies from the literature. Additionally,  the statistical analysis is not evident and the authors do not present any statistical data regarding the significant differences between the obtained results. The abstract should be rewritten and more explicitly present the main scope and results of the study. E.g. the sentence in lines 14-15 should be completed (carotenoids content in what?)

The abstract has been improved according to the reviewer’s suggestions. The present form of the abstract is quoted below:

“The paper presents the results of influence the light of different wavelength and pulsed electric field on the content of carotenoids. Seeds germination was carried out in the climatic chamber with phytotron system. The experiment was carried out under seven growing conditions differing in the light-emitting diodes (LED) wavelengths and using pulsed electric field (PEF) with different strength applied before sowing. Cultivation of the sprouts was carried out for 7 days at relative humidity 80% and 20 ± 1°C. The different light wavelengths were used during cultivation: white light (380-780 nm), UVA (340 nm), blue (440 nm), red (630 nm). In addition, the pulsed electric field (PEF) with three value of strength equal 1, 2.5 and 5 kV/cm respectively was applied to three series of sprouts before sowing. Sprouts treated with PEF were grown under white light (380-780 nm). The light exposure time for all experimental series of sprouts was 12/24h (12 hours light, 12 hours in the dark for 7 days). The lutein is the dominant carotenoid in germinating red clover seeds, the content of which varies from 743 mg/kg in sprouts grown in red light, 862 mg/kg in sprouts grown in UVA, to 888 mg/kg in blue light. Blue light in the cultivation of red clover sprouts (compared to white light cultivation) has the most beneficial effect on the increase of carotenoids content and amounts to 42% in the case of β-carotene, 19% lutein and 14% zeaxanthin. It confirms that modelling the content of carotenoids is possible using UVA and blue light (440nm) during seed cultivation. An increase in the content of β-carotene and lutein in red clover sprouts was obtained in comparison to the test with white light without PEF pre-treatment respectively by 8.5 and 6%. At the same time the decrease of 3.3% in the content of zeaxanthin was observed. Therefore, It can be concluded that PEF pre-treatment may increase mainly the content of β-carotene. “

The discussion according to the suggestions was expanded, numerical values appearing in the cited literature and new references were placed in the text. There is little research on carotenoids in red clover, and no such data in young seedlings was found in the available literature. Studies on the content of carotenoids in red clover sprouts are innovative. Therefore, the obtained results were compared with other sprouts, but belonging to the Fabeceae family. In addition, we've included a comparison of the red clover germination energy that can enrich this discussion.

New references mentioned in the Discussion section have been added to the list:

  • Pardo, G.P.;, Aguilar, C.H.; Martínez, F.R.; Pacheco, A.D.; Martínez, C.L.; Ortiz, E.M. High intensity led light in lettuce seed physiology (Lactuca sativa). Acta Agroph. 2013, 20(4), 665-677
  • Dziadek, K.; Kopec, A.; Drozdz, T.; Kielbasa, P.; Ostafin, M.; Bulski, K. ; Oziemblowski, M.; Effect of pulsed electric field treatment on shelf life and nutritional value of apple juice; Journal Of Food Science And Technology-Mysore 2019, 56 (3), 1184-1191
  • Jakubowski, T.; The effect of stimulation of seed potatoes (Solanum tuberosum L.) in the magnetic field on selected vegetation parameters of potato plants, Przeglad Elektrotechniczny 2020, 96 (1), 166-169
  • Sobol, Z.; Jakubowski, T.; The effect of storage duration and UV-C stimulation of potato tubers, and soaking of potato strips in water on the density of intermediates of French fries production, Przeglad Elektrotechniczny 2020, 96 (1), 242-245
  • Leong, S.Y.; Burritt, D.J.;  Oey, I..Electropriming of wheatgrass seeds using pulsed electric fields enhances antioxidant metabolism and the bioprotective capacity of wheatgrass shoots. Sci. Rep. 2016, 6, 1-13
  • Ahmed, Z.; Manzoor, M.F.; Ahmad, N. Impact of pulsed electric field treatments on the growth parameters of wheat seeds and nutritional properties of their wheat plantlets juice. Food Sci Nutr. 2020, 00:1–11
  • Evrendilek, G.A.; Karatas, B.; Uzuner, S.; Tanaso, I. Design and effectiveness of pulsed electricfields towards seed disinfection. J Sci Food Agric. 2019; 99, 3475–3480
  • Su, B.; Guo, J.; Nian, W.; Feng, H.; Wang., K. Early Growth Effects of Nanosecond Pulsed Electric Field (nsPEFs) Exposure on Haloxylon ammodendron. Plasma Process. Polym. 2015,12, 372–379
  • Fiutak, G.; Michalczyk, M.; Filipczak-Fiutak, M.; Fiedor, L.; Surówka, K. The impact of LED lighting on the yield, morphological structure and some bioactive components in alfalfa (Medicago sativaL.) sprouts. Food Chemi. 2019, 28, 553–58
  • Chauveau-Duriot, B.; Thomas, D.; Portelli, J., Doreau, M. Effect du mode de conservation sur la teneur en caroténoïdes des fourrages Carotenoid content in forages : variation during conservation. Renc. Rech. Ruminants, 2005, 12, 117
  • Zeb, A.; Hussain, A. Chemo-metric analysis of carotenoids, chlorophylls, and antioxidant activity of Trifolium hybridum. Heliyon. 2020, 6, e03195, 1-6
  • Cardinault, N.; Doreau, M.; Poncet, C.; Nozie`re, P. Digestion and absorption of carotenoids in sheep given fresh red clover. Animal Science. 2006, 82, 49–55
  • Mikhailov, A.L.;,. Timofeeva, O.A.; Ogorodnova, U.A.; Stepanov N.S. Comparative Analysis of Biologically Active Substances in Trifolium Pratense and Trifolium Repens Depending on the Growing Conditions. Journal of Environmental Treatment Techniques. 2019, 874-877

The manuscript has been enriched with the information about biomass increasing:

” In the case of crops of different varieties of lettuce sprouts, it was found that the increase in fresh weight increased in the case of using red or blue light in the grow process compared to crops under white light. The sprouts were irradiated for 12 hours and 12 hours left in the dark. The best effect of the growth of germ mass was obtained when the sprouts were exposed to 3 hours of blue light and in one case of the varieties it was observed that it increased by 23% [8]. On the other hand, in studies conducted by[25], the wheat seeds before cultivation were treated with 50 pulses of 6 kV/cm PEF, a significant increase in fresh weight was noted, which is positively correlated with PEF energy. “

The text about carotenoids and the data available in the literature has been placed in the Results and Discussion section:

Carotenoid content was studied in mature red clover plants for fodder grown in the field. Research shows that the main carotenoid is lutein, its content is 136 mg/kg DM (dry matter), and the total β-carotene content is 29 mg/kg DM [42]. In other studies, the composition of the carotenoids in red clover is different, and the content of lutein, β-carotene and zeaxanthin is much higher, respectively 237.7, 100.3; 91.7 mg/kg DM. It should be noted that in recent studies, clover after harvesting has been subjected to a consolidation process [43]

Other authors report that the content of β-carotene in red clover is about 200 mg/kg DM, and the content of this carotenoid in Trifolium repens varies from 300 to 730 mg/kg DM depending on the harvesting place. [44]

In the experiment where the subject was Alsike clover, also belonging to the genus Trifolium, the content of lutein and β-carotene were much higher and equaled to 208.9-243 mg/kg fresh weight and 35.4-123.4 mg/kg respectively [45].

The effect of various types of LED lighting on the content of carotenoids in alfalfa sprouts (also belonging to Fabaceae family) was investigated [46]. The combination of red and blue LED's used during cultivation increased lutein content from 82.6 to 108.2, and β-carotene from 26.6 to 44.6 mg/kg fresh mass of sprouts [46]. However, the research did not cover the whole sprout, only cotyledon. Increasing the intensity of blue light during the cultivation of beetroot (Beta vulgaris L.) and parsley (Petroselinum crispum Mill.) causes an increase in the content of carotenoids as compared to irradiation with red light. It has been shown that increasing the proportion of blue light intensity in these micorogreens results in an increase in lutein content from 103.8 to 118, 5 mg/kg of fresh beet mass, from 122.9 to 190.7 mg/kg of fresh parsley, and β-carotene, respectively from 0.09 to 0.87 and 0.54 to 0.86 mg/kg, and zeaxanthin from 1.39 to 3.20 and up to 0.84 -14.4, respectively [41].

Lefsrud in his research observed an increase in the content of lutein and β-carotene in kale both when illuminated with blue and red light. The maximum accumulation of lutein in fresh kale mass was found at 640nm and β-carotene at 440nm calculated on a fresh mass basis. However, when converted to dry mass, the maximum lutein content in kale also occurred at 440 nm. [1]

There are no reports on the content of carotenoids in red clover sprouts, as well on the impact of PEF on the content of carotenoids in the literature on the subject. Therefore, our research can only be compared with the unique research carried out by Ahmed at all [25] That group studied the effect of PEF on the content of plant dyes in wheat germ. A 35% increase in the content of carotenoids was observed in the PEF treatment with strength of 6kV/cm and 50 impulses compared to untreated sprouts [25]

In our experiment, no such significant effect of PEF treatment of red clover seeds on the content of carotenoids was found.

In the case of wheatgrass (Triticum aestivum L.) the usage of 0.5kV/cm PEF did not affect the growth of sprouts compared to the control ones. Increasing the PEF strength to 1.4 kV/cm had a positive effect on sprout growth, however, the 2 kV/cm PEF treatment of the tested seeds had an adverse effect and a lower growth was obtained than compared to the control [24]

The additional information about d=germination energy has been placed in the Results and Discussion section:

“A statistically significant increase in germination energy was found after 4 days using a field of energy 240J (12 kV) for parsley and 960J (12kV, the number of pulses was changed) for parsley, winter wheat, winter barley, lettuce, tomato and garden rocket [37].

In other studies, germination energy measured after 24 h was positively correlated with the increase in intensity (energy of pulse electric field) and in the case of 6kV/cm PEF with 50 pulses it was 92% compared with 84% for non-treated sprouts [25].

For Haloxsylon ammodendron seeds, it has been observed that germination energy increases from 72.9% to 90.3 and 98.0% in the cases of using the strengths of 10 and 20 kV/cm respectively [38].”

 All numbers of tables and references have been corrected.

Attached please find the improved and upgraded text of our article.

Reviewer 3 Report

The topic of the study – the impact of pulsed electric field and Light emitting diode lighting on carotenoid contents in red clover sprouts – is interesting and novel for publication in Applied sciences. Both tools have enough attention for application in practice and both are proved to be efficient at experimental level. The topic is actual and novel enough, however, the novelty of the study was not disclosed in the manuscript (see comments below). Also, the overall quality of the manuscript is weak. The experimental design is not presented clearly and lacks of various information (see comments below). The data presented (The red clover sprout weight and carotenoid contents) is interesting, however, the amount of the experimental information presented here is not sufficient. Maybe authors have other supplementing information, like other growth parameters (height, leaf area), describing the impact of investigated parameters on the morphology of sprouts, antioxidant activity of the material or other parameters proving the impact of light or/and electric field? Result presentation, as well as discussion is not convincing and the conclusions are not concrete enough.

Abstract – how was the electric field generated? What was the intensity of LED light applied? Please include the more concrete results, explaining the impact of lighting and the impact of PFE

Introduction – Lot’s of information about carotenoids, however only minor information on how the carotenoid contents in plants can be manipulated. More attention should be payed for LED and PFE impacts on plants, therefore explaining the novelty of the idea from the experimental side, not only stress the fact, that red clover sprouts were not investigated from that point. Only once, in method section, L140 “electroporation” term was mentioned. Why this term/effect was not explored in the text?

Materials and methods. Experimental design is not presented properly. Was the PFE applied on the seeds, or already on the sprouts? Please explain in detail. For the light exposure, L157, was the 12 h ligt/24 h dark period applied? Please explain. What was the intensity (PPFD) of light applied for each wavelength? What was the light source?

What was the size and number of containers per treatment? Was any substrate used? How was the experiment organized? 49 treatments, but how the treatments were organized? How experimental replications were arranged? How the plant material was collected (from replications).

Results and discussion

Information in lines 191-222 is more suitable for introduction and materials and methods

L223 – how the mass was obtained? It is not explained in M&M section, how many plants and how were measured?

Figure 1 (L242) has no anova results presented. Is there a mass of single sprout? Or what?

Table 1 also represents the mass of sprouts, but the data does not match the one, presented in fig. 1. Please explain the source of the numbers in both figures and revise, if both objects are necessary, if they present the supplementary information?

Figure 2 – 3 independent measurements, what are they? Analytical replications, 3 extracts form single treatment aor biological replications of the treatments?

L258 – documented effect? Please give references for these effects.

Conclusions – most of the text is more suitable for discussion. For the conclusions, it would be expected for more concrete facts and trends.

Author Response

Dear Reviewer,

We would like to thank you very much for your time you spent for correcting our paper and suggestions. They have helped us to improve the paper.

Below we have written answers for your comments. All of them are placed directly under the comments and they are highlighted in the text :

The topic of the study – the impact of pulsed electric field and Light emitting diode lighting on carotenoid contents in red clover sprouts – is interesting and novel for publication in Applied sciences. Both tools have enough attention for application in practice and both are proved to be efficient at experimental level. The topic is actual and novel enough, however, the novelty of the study was not disclosed in the manuscript (see comments below). Also, the overall quality of the manuscript is weak. The experimental design is not presented clearly and lacks of various information (see comments below). The data presented (The red clover sprout weight and carotenoid contents) is interesting, however, the amount of the experimental information presented here is not sufficient. Maybe authors have other supplementing information, like other growth parameters (height, leaf area), describing the impact of investigated parameters on the morphology of sprouts, antioxidant activity of the material or other parameters proving the impact of light or/and electric field? Result presentation, as well as discussion is not convincing and the conclusions are not concrete enough.

Abstract – how was the electric field generated? What was the intensity of LED light applied? Please include the more concrete results, explaining the impact of lighting and the impact of PFE

The abstract has been improved according to the reviewer’s suggestions. The present form of the abstract is quoted below:

“The paper presents the results of influence the light of different wavelength and pulsed electric field on the content of carotenoids. Seeds germination was carried out in the climatic chamber with phytotron system. The experiment was carried out under seven growing conditions differing in the light-emitting diodes (LED) wavelengths and using pulsed electric field (PEF) with different strength applied before sowing. Cultivation of the sprouts was carried out for 7 days at relative humidity 80% and 20 ± 1°C. The different light wavelengths were used during cultivation: white light (380-780 nm), UVA (340 nm), blue (440 nm), red (630 nm). In addition, the pulsed electric field (PEF) with three value of strength equal 1, 2.5 and 5 kV/cm respectively was applied to three series of sprouts before sowing. Sprouts treated with PEF were grown under white light (380-780 nm). The light exposure time for all experimental series of sprouts was 12/24h (12 hours light, 12 hours in the dark for 7 days). The lutein is the dominant carotenoid in germinating red clover seeds, the content of which varies from 743 mg/kg in sprouts grown in red light, 862 mg/kg in sprouts grown in UVA, to 888 mg/kg in blue light. Blue light in the cultivation of red clover sprouts (compared to white light cultivation) has the most beneficial effect on the increase of carotenoids content and amounts to 42% in the case of β-carotene, 19% lutein and 14% zeaxanthin. It confirms that modelling the content of carotenoids is possible using UVA and blue light (440nm) during seed cultivation. An increase in the content of β-carotene and lutein in red clover sprouts was obtained in comparison to the test with white light without PEF pre-treatment respectively by 8.5 and 6%. At the same time the decrease of 3.3% in the content of zeaxanthin was observed. Therefore, It can be concluded that PEF pre-treatment may increase mainly the content of β-carotene.”

Introduction – Lot’s of information about carotenoids, however only minor information on how the carotenoid contents in plants can be manipulated. More attention should be payed for LED and PFE impacts on plants, therefore explaining the novelty of the idea from the experimental side, not only stress the fact, that red clover sprouts were not investigated from that point. Only once, in method section, L140 “electroporation” term was mentioned. Why this term/effect was not explored in the text?

The introduction has been supplemented with the information about the effect of LED and PFE on plants. The term "electroporation" L140 has been explained.

The following text has been placed in the manuscript:

In the Introduction section:

“Plant growth, including germ and the profile of biologically active compounds depends on the genotype, type of exposure (monochrome, combined or white light), its intensity and time [8]”

“It is believed that PEF can potentially be used to control and optimize the process of sprout growth and modify its composition, in particular nutritional values and bioactive ingredients [24 ]. PEF can also effectively stimulate germ growth, positively affect metabolism and nutrient content [24,25].

The effect of PEF on germination depends on the type of plant and the strength of the used field, while on some species such as marigold tomato or radish no significant effect was observed. In the case of lentil a 50% increase in germination rate was observed. The effect on changes in germination rate and germination rate induced by PEF is probably associated with changes in metabolism of amino acids occurring in seeds [25] while PEF induces electroporation causing increased membrane permeability. Electroporation is a reversible process, however, when too large parameters of the PEF process are used, irreversible changes in the structure of the cell membrane can occur. [26]. Depending on the duration of the pulses, the number of pulses and the strength of the electric field, the cell membrane may even be destroyed, i.e. irreversible electroporation [25]. 

Red clover Trifolium pratense L. is an interesting plant. In the world, it is the most often known as a feed plant because it is used for feed. There are many publications on bioactive compounds and their variability resulting from the cultivation of a mature plant. Many pharmaceutical preparations and dietary supplements [27,28] are also obtained on the basis of this plant. There is little research on carotenoids in red clover, and no such data was found in the available literature. Research and modelling of carotenoids content in red clover sprouts are innovative.”

The additional references have been placed:

  • Pardo, G.P.; Aguilar, C.H.; Martínez, F.R.; Pacheco, A.D.; Martínez, C.L.; Ortiz, E.M. High intensity led light in lettuce seed physiology (Lactuca sativa). Acta Agroph. 2013, 20(4), 665-677
  • Leong, S.Y.; Burritt, D.J.;  Oey, I..Electropriming of wheatgrass seeds using pulsed electric fields enhances antioxidant metabolism and the bioprotective capacity of wheatgrass shoots. Sci. Rep. 2016, 6, 1-13
  • Ahmed, Z.; Manzoor, M.F.; Ahmad, N. Impact of pulsed electric field treatments on the growth parameters of wheat seeds and nutritional properties of their wheat plantlets juice. Food Sci Nutr. 2020, 00:1–11
  • Saulis, G. Electroporation of Cell Membranes: The Fundamental Effects of Pulsed Electric Fields in Food Processing. Food Eng Rev. 2010, 2, 52–73
  • Bemis, D.L; Capodice, J.L; Costello, J.E.; Vorys, G.C.; Katz, A.E.; Buttyan, R. The Use of Herbal and Over the counter Dietary Supplements for the Prevention of Prostate Cancer. Oncol. Rep. 2006, 8, 228–236
  • Tundis, R.; Marrelli,M.; Conforti, F.; Tenuta, M.C.; Bonesi, M.; Menichini, F.; Loizzo, M. Trifolium pratense and repens (Leguminosae): Edible Flower Extracts as Functional Ingredients. Foods 2015, 4, 338-348

Materials and methods. Experimental design is not presented properly. Was the PFE applied on the seeds, or already on the sprouts? Please explain in detail. For the light exposure, L157, was the 12 h ligt/24 h dark period applied? Please explain. What was the intensity (PPFD) of light applied for each wavelength? What was the light source?

According to the reviewer’s suggestions the additional information has been placed in the Materials and Methods section:

“PEF was applied to red clover seeds before germination. “

“During cultivation, lighting was used repeatedly for 12 hours and after that period the plants were stored 12 hours in the dark.”

“The LEDs which emitted the light with different wavelengths were used as the light sources in the process of sprouts growing.”

“A photosynthetic photon flux density (PPFD) of 150 ± 5 μmol m-2 s-1 was maintained.”

“Seeds after PEF treatment were stored for 14 days at room temperature (T = 20 ° C) and then grown.”

What was the size and number of containers per treatment? Was any substrate used? How was the experiment organized? 49 treatments, but how the treatments were organized? How experimental replications were arranged? How the plant material was collected (from replications).

The designed phytotron chamber was used for the experiments. Seeds were placed in the containers, 3 crops were run parallel in each type of test. In the research, 3 containers were used for each type of experiment. The containers were 5cm x 8cm. During seed cultivation, no substrate was used. Whole sprouts were harvested by hand on the last day of cultivation.

The additional information has been placed in the text:

“Sprouting was conducted in triplicate for each treatment (3 containers for each experiment).”

and

“Sprouts were harvested manually every 24 hours from the day of sowing. Analyses from the average sample from each container (3 containers for each treatment) were performed in one test samples taken from each container. After each harvest, the weight of harvested sprouts was determined. The content of carotenoids was determined after 7 days of red clover sprouts growing.”

Results and discussion

Information in lines 191-222 is more suitable for introduction and materials and methods

These are valuable tips. The relevant fragments have been moved to the theoretical introduction and methodology section.

L223 – how the mass was obtained? It is not explained in M&M section, how many plants and how were measured?

According to the reviewer’s suggestion, the additional information has been placed in subsection Sprouts Cultivation:

“Red clover sprouts were weighed immediately after harvesting, on the RADWAG analytical PS 06.R2 The obtained germs of the sprouts are the average of three samples of 5 g seeds each. From the time the seeds were sown, the samples were weighed every 24 hours.”

Figure 1 (L242) has no anova results presented. Is there a mass of single sprout? Or what?

Figure 1 shows the increase in fresh weight of sprouts obtained from 5g seeds (received in a 24-hour time regime) with standard deviation.

Table 1 shows the increase in fresh weight on the last day of the experiment per 100g of seeds. The imprecise description of the table was supplemented and ANOVA results were placed in the manuscript. This table is intended to indicate statistical differences.

In addition, we've included a comparison of the red clover germination energy that can enrich the discussion.

The additional subsection has been added to the manuscript:

“2.4 Germination energy

For each cultivation variant, tree hundred seeds (red clover and red clover after PEF treatment) were collected. They were grown in the separate molds made of PP plastic, dedicated to conducting the germination process. During the research, the percentage of healthy, correctly germinated seeds was determined. During the cultivation, seed viability indexes were determined: energy of red clover seeds.

Germination energy (GE) was determined as the percentage of seeds that germinated during the first 4 days (96 h).”

And

“The germination energy of red clover seeds (Table 1) in the case of PEF 5kV/cm (in the process of seed pretreatment before sowing) was the highest and differed statistically in comparison to other treatments. The use of monochromatic LED sources and white light with the addition of 1 and 2.5 kV/cm PEF did not affect germination energy compared to white light crops (no statistically significant differences were found).”

Table 1 also represents the mass of sprouts, but the data does not match the one, presented in fig. 1. Please explain the source of the numbers in both figures and revise, if both objects are necessary, if they present the supplementary information?

Table 1 shows the weight of sprouts on the last day of the experiment per 100g of seeds. The inaccurate description of the table was supplemented and ANOVA results included, and germination energy (GE) values added. This table is to indicate statistical differences.

The results of the statistical analysis regarding statistically significant differences are presented in the Tables 2-5.

Table 1. Germination energy of given per 100 pieces of seeds and the final mass of the samples after their cultivation in the different growth conditions, recalculated to the initial weight of 100g sprouts. Values are presented as: mean ± SD.

light

germination energy [%]*

(n = 4)

final mass [g]**

(n = 3)

UVA (340 nm)

89.3 ± 2.9

600.8 ± 36.6

blue (440 nm)

88.5 ± 2.1

629.0 ± 30.2

red (630 nm)

88.5 ± 2.3

778.6 ± 34.3

white (380-780 nm)

89.0 ± 2.3

669.5 ± 28.1

white light and pulsed electric field pretreatment

field strength

1.0 kV/cm

88.8 ± 2.3

672.0 ± 21.7

2.5 kV/cm

90.8 ± 2.9

692.0 ± 20.8

5.0 kV/cm

94.0 ± 3.0

710.4 ± 30.5

* The performed analysis of variance does not allow for the rejection of the hypothesis of equality of means (F=1.82, p=0.143).Kruskal-Wallis test was conducted to examine the differences on germination energy according to the treatment taken. No significant differences (Chi square =10.91, p = 0.09) were found among all the seven growing conditions.

**  Statistically significant differences are presented in the table 2.

Table 2. P values of Tukey post hoc tests after one-way ANOVA for final mass of sprouts with treatments as fixed factors, F=11,50, significance level p< 0.05. Statistically significant differences are bolded

treatment

UVA

blue

red

white

PEF 1

PEF 2.5

UVA

blue

0.8931

red

0.0002

0.0005

white

0.1299

0.6335

0.0066

PEF 1

0.1094

0.5735

0.0079

1.0000

PEF 2.5

0.0261

0.1965

0.0342

0.9637

0.9796

PEF 5

0.0063

0.0522

0.1345

0.6254

0.6850

0.9827

Table 3. P values of Tukey post hoc tests after one-way ANOVA for β-carotene with treatments as fixed factors, F=30.84, significance level p< 0.05. Statistically significant differences are bolded

treatment

UVA

blue

red

white

PEF 1

PEF 2.5

UVA

blue

0.2244

red

0.0056

0.0002

white

0.0002

0.0002

0.0866

PEF 1

0.0002

0.0002

0.1591

0.9997

PEF 2.5

0.0004

0.0002

0.6018

0.8137

0.9470

PEF 5

0.0010

0.0002

0.9321

0.4290

0.6354

0.9911

Table 4. P values of Tukey post hoc tests after one-way ANOVA for lutein with treatments as fixed factors, F=21.10, significance level p< 0.05. Statistically significant differences are bolded.

treatment

UVA

blue

red

white

PEF 1

PEF 2.5

UVA

blue

0.7675

red

0.0003

0.0002

white

0.0003

0.0002

1.0000

PEF 1

0.0006

0.0002

0.9915

0.9966

PEF 2.5

0.0133

0.0011

0.2138

0.2511

0.5309

PEF 5

0.0156

0.0013

0.1859

0.2194

0.4801

1.0000

Table 5. P values of Tukey post hoc tests after one-way ANOVA for zeaxanthin with treatments as fixed factors, F=11,07, significance level p< 0.05 Statistically significant differences are bolded.

treatment

UVA

blue

red

white

PEF 1

PEF 2.5

UVA

blue

0.0015

red

0.2638

0.0002

white

0.6442

0.0287

0.0151

PEF 1

0.8147

0.0163

0.0266

0.9999

PEF 2.5

0.4240

0.0575

0.0075

0.9996

0.9903

PEF 5

0.9842

0.0057

0.0755

0.9669

0.9963

0.8470

Figure 2 – 3 independent measurements, what are they? Analytical replications, 3 extracts form single treatment or biological replications of the treatments?

3 replications of the treatments were done. Seeds were seeded into 3 containers for each type of lighting and PEF pretreatment. The analyses of the average sample from each container were performed in one replicate.

The following text has been added into the 2.3 Sprouts cultivation section :

“Sprouting was conducted in triplicate for each treatment (3 containers for each experiment).”

L258 – documented effect? Please give references for these effects.

The effect was documented and the literature was supplemented as recommended. The following references were added:

  • Samuolienė, G.; Viršilė, A.; Brazaitytė, A.; Jankauskienė, J.; Sakalauskienė, S.; Vaštakaitė, V.; Novičkovas, S.; Viškelienė, A.; Sasnauskas, A.; Duchovskis, P. Blue light dosage affects carotenoids and tocopherols in microgreens. Food Chem. 2017, 228, 50-56.
  • Frede, K.; Schreiner, M.; Baldermann, S. Light quality-induced changes of carotenoid composition in pak choi Brassica rapassp. chinensis. Journal of Photochemistry & Photobiology, B: Biology. 2019, 193, 18–30
  • Folta, K.M.; Maruhnich, S.A. Green light: a signal to slow down or stop. Exp. Bot., 2007, 58, 12, 3099–3111
  • Li, Y.; Zheng, Y., Liu, H. Effect of supplemental blue light intensity on the growth and quality of Chinese kale. Hortic. Environ. Biotechnol. 2017, 60, 49–5

Conclusions – most of the text is more suitable for discussion. For the conclusions, it would be expected for more concrete facts and trends.

The reviewer’s suggestion has been taken into account and the Conclusions section has been extended with the following information:

“The dominant carotenoid in germinating red clover seeds is lutein, the content of which varies from 743 mg/kg in sprouts grown in red light, 862 mg/kg in sprouts grown in UVA, to 888 mg/kg in blue light. UVA and blue light are most preferred for obtaining carotenoid dyes. the highest content of the total value of tested carotenoids (β-carotene, lutein and zeaxanthin) in red clover sprouts was obtained, respectively 1750 mg/kg (kg of dry weight) in the case of UVA irradiation and 1892 mg/kg using blue irradiation. “

 All numbers of tables and references have been corrected.

Attached please find the improved and upgraded text of our article.

Round 2

Reviewer 2 Report

The authors addressed sufficiently the reviewers' comments and the manuscript can be accepted for publication.

Author Response

Dear Reviewer,

We would like to support our work. The corrected version of manuscript is resubmitted.